# Back to School: An Updated Evaluation of the Effectiveness of a Long-Term Trap-Neuter-Return Program on a University’s Free-Roaming Cat Population

**DOI:** 10.3390/ani9100768

**Published:** 2019-10-08

**Authors:** Daniel D. Spehar, Peter J. Wolf

**Affiliations:** 1Independent Researcher, 4758 Ridge Road, #409, Cleveland, OH 44144, USA; danspehar9@gmail.com; 2Best Friends Animal Society, 5001 Angel Canyon Road, Kanab, UT 84741, USA

**Keywords:** trap-neuter-return (TNR), feral cats, free-roaming cats, community cats, non-lethal management, population reductions

## Abstract

**Simple Summary:**

Since the early 1990s, the use of trap-neuter-return (TNR) as a humane alternative to the lethal management of stray and feral cats (also known as community cats) has expanded in the United States. Over this time, numerous studies have pointed to the effectiveness of TNR at reducing community cat numbers, although many of these investigations have covered relatively short time periods (3 years or less). A seminal paper by Levy et al. in 2003, documented a significant decline in community cat numbers on the campus of the University of Central Florida (UCF) over a 11-year period. Since 2017, a series of peer-reviewed articles have described other examples of long-term reductions in community cat numbers associated with TNR. The present study adds to this growing body of evidence by revealing the extent to which the results first reported by Levy et al. have been sustained over a subsequent 17-year period. After a total of 28 years, 10 (5%) of 204 total cats enrolled in the UCF TNR program, remain on campus and the campus community cat population has declined by 85% from the completion of an initial census in 1996 to 2019.

**Abstract:**

A growing body of evidence indicates that trap-neuter-return (TNR) is not only effective at reducing community cat numbers, but that such reductions are sustainable over extended periods. Recently, a series of peer-reviewed articles documenting long-term declines in community cat populations associated with TNR have been published. The present study adds to this pool of evidence by updating and reexamining results reported from the campus of the University of Central Florida (UCF) in 2003 by Levy et al. From 1991 to 2019, a total of 204 cats were enrolled in a volunteer-run TNR program on the university grounds; 10 cats (5%) remained on site at the conclusion of the present study. The campus community cat population declined by 85% between 1996, the year an initial census (indicating the presence of 68 cats) was completed, and 2019. In addition, 11 of 16 total colonies were eliminated over a 28-year period. These results occurred despite significant growth in enrollment at UCF over the same time frame, which suggests that with sufficient ongoing management of colony sites, declines in community cat populations associated with TNR are sustainable over long periods and under varying conditions.

## 1. Introduction

For decades much controversy has surrounded the management of stray and feral cats (henceforth referred to as community cats). Among the key factors prompting policy makers to take action are concerns over wildlife and public health impacts [1,2,3], nuisance complaints [4,5], and animal welfare concerns [6,7,8]. Since the early 1990s, the use of trap-neuter-return (TNR) as a humane alternative to the lethal management of community cats has proliferated in the United States. By 2006, several studies from the U.S. and elsewhere had documented the impacts of TNR on community cat populations, including declines in population size [9,10,11,12,13,14] and reductions [15] or elimination [16] of kitten births. Yet, only three of these studies [11,13,14] encompass time periods of more than three years. One- [17] or two-year [18] studies of TNR, on the other hand, have documented population increases, similar to the short-term increases documented following the implementation of TNR in at least one long-term study [14]. More long-term studies would allow for a better assessment of the impacts of TNR programs.

Recently, a number of TNR studies documenting long-term reductions in community cat populations have been published: nine years by Swarbrick et al. (2018) [19], 10 years by Spehar and Wolf (2018) [20], 14 years by Kreisler et al. (2019) [21], and 17 years by Spehar and Wolf (2017) [22]. The present study builds upon this recently published body of evidence by updating and reexamining the results, as first reported by Levy et al. [11] in 2003, of a TNR program that has now been carried out on the campus of the University of Central Florida (UCF) over a 28-year period (1991 to 2019). Here, we assess the extent to which the reductions in population size and changes in population dynamics observed on the UCF campus over a 11-year period ending in 2002 persisted over an additional 17 years.

## 2. Materials and Methods

### 2.1. Location

The University of Central Florida is located 21 km east of downtown Orlando, Florida, and covers 573 hectares. The campus population of students and staff has risen from an estimated 38,000 in 2002 to over 68,000 students and 13,000 employees in 2018 [23]. Increased enrollment at the university has led to considerable and ongoing development of the campus landscape (Figure 1), in the form of new classroom buildings, dormitories, and sports/entertainment facilities; however, a large portion of the outer perimeter of the campus remains wooded, including a 32-hectare arboretum on its eastern edge [24]. Nearly the entire campus is included in the TNR program area.

### 2.2. Background

Several colonies of cats formed near food service buildings and dormitories soon after the university opened in 1968. Over the ensuing decades, the number of colonies grew and cats were periodically trapped and euthanized in response to nuisance complaints. A concerted management program consisting of trapping, sterilization, vaccination, and return of cats to campus locations or adoption began in 1991.

The program protocol called for cats to be trapped in humane box traps, transported by Friends of Campus Cats to a veterinary facility (over the years these included private veterinary practices, in-house clinics at the local humane society, and Orange County Animal Services, as well as two local high quality/high volume sterilization clinics used presently) to be sterilized and vaccinated for rabies and viral rhinotracheitis/calicivirus/panleukopenia (FVRCP). As practicable, follow-up rabies vaccines have been administered to cats who have remained on campus for a number of years. Cats were also ear-tipped and received minor medical care (e.g., flea/ear mite treatment, deworming, and wound care), as warranted and resources allowed. Routine feline immunodeficiency virus/feline leukemia virus (FIV/FeLV) testing of selected cats (i.e., those appearing ill and mature males) was part of the standard program protocol until 2000. Cats testing positive were euthanized. The last cat euthanized for a positive FIV test was in March of 2000, while the last cat euthanized for a positive FeLV test was in September of the same year.

A census of the campus cat population, began upon inception of the program in 1991 and completed in 1996, was conducted by four members of Friends of Campus Cats. The number of cats residing on campus has been monitored on an ongoing basis since completion of the initial census. During daily feeding sessions, caretakers noted novel or unusual circumstances at colony sites, including suspected health problems among the cats and changes in attendance (absences and new arrivals). The maximum number of feeding stations (Figure 2) installed across the program area was 12. The feeding stations were spaced an average of 100 m apart. The number of colony caretakers varied, depending on need, from 7 to 12.

### 2.3. Data Collection

Cats were recorded and their presence tracked as they were discovered on the UCF campus. Each cat was assigned a name and tracked by colony affiliation on a Microsoft Excel spreadsheet. The enrollment date, description (as well as a photograph), sex, age category (adult or kitten), perceived level of socialization at first appearance (socialized or feral), date neutered/neuter status, departure date (if applicable), and final outcome (if applicable) for each cat was documented. Kittens were defined as cats estimated to be six months of age or younger. Level of socialization was determined at enrollment and was not revised to reflect changes over time, even when cats subsequently became more social or friendly to specific individuals (e.g., colony caretakers).

Spreadsheets tracking the data listed above, updated through March of 2019, were analyzed as part of the present study. End-point data were compared to results of the initial census and to published results from the conclusion of the initial observation period in 2002. Semi-structured interviews were conducted with two of the founders of Friends of Campus Cats, who were also co-authors of the 2003 paper, in order to provide context and fill information gaps.

### 2.4. Data Analysis

Consistent with what was reported by Levy et al. [11], descriptive statistics were calculated for data regarding population variables. End-point results were incorporated for the time period of January 1991 through March 2019 (“combined observation period”). Results from the initial census in 1996 and from the end of the “initial observation period” (1991–2002) were compared to the results at the conclusion of the “follow-up observation period” (2002–2019) in order to assess the extent to which impacts of TNR on the population of community cats residing on the UCF campus were sustained over an extended length of time.

## 3. Results

A total of 204 cats were recorded on campus during the combined 28-year reporting period ending in March of 2019; 49 (24%) of the cats arrived on campus after the end of the initial observation period in 2002, at which time 155 (76%) cats had been recorded. At the end of the combined observation period, the total number of cats recorded on campus consisted of 43 (21%) socialized cats and 161 (79%) unsocialized feral cats (at first appearance); 111 (54%) adults and 93 (46%) kittens (age category determined at enrollment); and 114 (56%) males and 90 (44%) females (Table 1). From the end of the initial observation period to the conclusion of the combined observation period, the number of cats already neutered at the time of initial capture increased from 7 (5%) to 14 (7%), including 9 socialized adults, 4 feral adults, and 1 feral kitten. Overall, 174 (85%) of the 204 total enrolled cats were sterilized, 160 (78%) as part of the program, while 11 (5%) were euthanized for serious health concerns without being sterilized, 9 (4%) were considered too young to undergo sterilization surgery prior to adoption, 7 (3%) disappeared before they could be trapped, and 3 (1%) were adopted before being sterilized for reasons other than age.

The median number of cats enrolled in the program during the initial observation period was 11 per year (range, 7–35) compared to the median number of 2.5 enrolled per year (range, 0–6) during the follow-up observation period. The amount of time between enrollment in the program and sterilization surgery declined from an average of 1 year (median 0.5; range, 0–9.9) for the initial observation period to an average of 0.1 years (median 0; range, 0–1.5) for the follow-up observation period (Figure 3).

At the end of the combined observation period, 10 (5%) of the 204 total cats remained on campus; 23 (15%) of the original 155 cats remained on campus at the end of the initial observation period, so this change documented between observation periods represents a 57% decline in population over the intervening 17 years. The cats remaining on campus at the end of the combined observation period had resided there for a median of 3 years (range, 0–10); no cats continued to remain on campus from the end of the initial observation period. When the number of cats remaining on campus at the conclusion of the combined observation period was compared to the 68 cats recorded on campus at the end of the initial census, which concluded in 1996, an 85% decline in population size occurred over 23 years. All 10 of the cats who remained on campus in 2019 had been sterilized.

In addition to the 10 cats remaining on site at the end of the combined observation period, 92 (45%) cats had been adopted, 49 (24%) disappeared, 23 (11%) were euthanized, 17 (8%) were known to have died, 10 (5%) relocated to one of the wooded areas that surround much of the campus (status unknown), and 3 (1%) were confirmed to have moved to a location on the outskirts of campus outside of the program area (Table 2).

Overall, on an absolute basis, the number of cats adopted from the campus TNR program between 2002 (end of the initial observation period) and 2019 (end of the combined observation period) increased from 73 to 92 (Table 3); however, adoptions as a percentage of the total number of cats enrolled declined from 47% to 45%. Both in absolute terms and as a percentage of the total, the number of socialized adults, feral adults, and feral kittens adopted from the program increased, while the number of socialized kittens adopted was unchanged (Table 3). Notably, 39 of the 49 (80%) cats enrolled in the program between 2002 and 2019 were feral adults, as were 11 of the 19 (58%) cats adopted during the same period. Consistent with what was reported by Levy et al. [3] at the end of the initial observation period, results for the combined observation period showed that socialized adults and kittens were adopted soon after capture and neutering (median of 0.1 years; range, 0–5.8), whereas feral kittens (median of 0.5 years; range, 0–11.6) and adults (median of 3.7 years; range, 0–16.3) remained on site longer before adoption; overall, cats who were adopted spent a median of 0.3 years (range, 0–16.3) on campus.

Cats not adopted within six months of enrollment (152), regardless of their ultimate disposition, spent an average of 4.2 years (median 3.3; range, 0–18.3) on campus; 91% of such cats were feral, including 34 of 40 who were adopted on an average of 5.1 years (median 4.9; range, 0.7–16.3) after arriving on campus.

Between the end of the initial observation period in 2002 and the combined observation period in 2019, the total number of cats known to have died after being returned increased from 10 (6%) to 17 (8%); eight were killed by automobiles and nine died of unknown causes. Such cats resided on campus for a median of 3.9 years (range, 0.3–8.3). The number of cats euthanized increased from 17 (11%) to 23 (11%); nine were euthanized due to serious illness, two after a traumatic injury, and 12 after testing positive for FIV (10) or FeLV (2). Cats who were euthanized lived for a median of 4.3 years (range, 0–13.6) on campus.

The number of cats who disappeared rose from 23 (15%) to 49 (24%). Cats who disappeared resided on campus for a median of 3.1 years (range, 0–15.2). During the follow-up observation period, one additional cat moved “to woods.” This cat was not observed subsequent to relocation to the wooded research park located adjacent to the southern edge of the UCF campus. Cats last seen in the woods resided on campus for a median of 0.3 years (range, 0–9). All but one cat (58/59), who either disappeared or moved to the woods, were feral.

Since the end of the initial observation period, three cats were confirmed to have joined a colony on the periphery of the campus grounds, outside of the program area, requiring the creation of a new category of disposition (“to out of area”) as listed in Table 2. Over the years, TNR has been practiced intermittently at locations adjacent to the UCF program area; the three cats in question relocated to such an area.

In all, 16 discrete colonies were identified (1 colony, consisting of a single cat, that was renamed after being moved a short distance due to a building demolition was counted only once); in addition, a varying number of cats roamed with an apparent attachment to no particular colony (collectively categorized as cats of “no fixed colony,” NFC). The maximum number of cats in each of the 16 colonies ranged from 1 to 25; the number of NFC cats ranged from 0 to 11. Overall, 11 of the 16 colonies were eliminated, three had the same population (1) at the end of the combined observation period as at enrollment, and two colonies increased in size (one colony from one to two cats and the other colony from two to three cats); two NFC cats were documented at the inception of the program in 1991; and two (different) NFC cats were documented in 2019.

As of 2002, Levy et al. reported that 3 of 11 colonies had been eliminated and that “despite the presence of cats for 7 to 9 years before colonies were disbanded and the ongoing availability of food, these colonies have not been reestablished by new arrivals” [11]. As of the 2019 follow-up, the three colonies described by Levy et al. remained disbanded. Cats were eliminated from nine additional colony sites during the follow-up observation period. Feeding stations were removed from sites where colonies were eliminated. At only one of these sites was a feeding station returned after it had been removed. In this case, the feeding station was put back six years after removal when two feral immigrants were trapped, sterilized, and returned; one of these cats remained on site at the 2019 follow-up.

As observed by Levy et al. [11], not all cats remained in the same colony for the duration of their time on campus. Of the 204 total cats, 24 (12%) changed colony affiliation (location) at least once: 17 cats moved only once, four cats moved a total of two times, and three cats moved three times. All but one of these cats arrived on campus during the initial observation period. In addition, a single colony consisting of one cat (who had been on campus for 9.4 years) was moved and renamed after the demolition of a building; that cat remained at the new colony location (for an additional 0.6 years) at the 2019 follow-up. Feral cats comprised 23 out of 24 cats who changed colony location; 15 of the feral colony changers were enrolled in the program as kittens. A total of 41 cats (20%), 22 male and 19 female, spent some portion of their time on campus in the NFC group (median 0.3 years; range, 0–9.1); for 30 cats, this comprised the entirety of their time on campus (median 0.1 years, range 0–9.1). Overall, the 24 cats (13 male and 11 female) who changed colony affiliation at least once (including those who spent some time in the NFC group) spent a median of 7.9 years (range, 1.8–18.3) on campus, including 3.1 years (range, 0.1–6) in their original colony and 3.4 years (range, 0.2–15.2) in a second colony. A total of seven cats spent a median of 1.5 years (range, 0.3–3) in a third colony and three cats resided in a fourth colony for a median of 2.8 years (range, 2.6–3.3) (Figure 4). All cats who changed colonies two or more times spent time in the NFC group; two of the three cats who changed colony affiliation three times ended their time on campus in the NFC group.

## 4. Discussion

The purpose of the present study was to determine the extent to which results of a long-term TNR program, as reported in a seminal work by Levy et al. [11] in 2003, were sustained over a subsequent 17-year period. The initial study, which reported on a volunteer-based TNR program located on a large university campus, covered a 11-year period and was the first investigation of its kind to span more than three years. The foregoing analysis of a combined 28 years of data reveals that the initial impacts of the UCF campus TNR program on population size and dynamics have not only persisted over the intervening years, but in many respects have grown.

### 4.1. Population Size

Levy et al. [11] reported a 66% decline in the number of free-roaming cats living on the UCF campus over six years based upon an initial census completed in 1996; in 2002, the cats who continued to reside on the university grounds was 15% of the total number of cats enrolled in the program after its inception in 1991. At the time of follow-up in 2019, 49 additional cats had been enrolled in the program since 2002 with 10 cats remaining on campus (as stated above, an additional three individuals were known to have moved outside of the program area where they continued to live as community cats), just 5% of the total number enrolled over the entirety of the program, and an 85% decline from the number of cats present in 1996 (Figure 5). The 85% decline in population over a 23-year period exceeds what has been documented as part of three long-term studies of shorter duration: on a university campus in Sydney, Australia (78% reduction in population size over nine years) [19]; on the grounds of a private residential community in Key Largo, Florida (55% reduction in population size over 14 years) [21]; and in an urban Chicago, Illinois, neighborhood (54% mean reduction in colony size over 4 to 10 years, and a 41% decline in total population size from 75 cats at program entry to 44 at the end of the observation period) [20].

Similarly, the percentage of the total number of cats enrolled in the program who remained on site at the end of the combined study period (5%) was smaller than what was observed at the other three locations (Table 4). Perhaps the results attained on the UCF campus over a 28-year period most closely resemble what occurred contemporaneously on the waterfront in Newburyport, Massachusetts, where a long-term TNR program eliminated a population of an estimated 300 resident cats over a 17-year period; due to ongoing management, the results attained there have persisted over the ensuing decade [22]. Moreover, Natoli et al. recently reported that in Rome, Italy, an 18-year TNR program, including the monitoring of colonies by registered caretakers, has reduced the problem of immigration into colonies by abandoned cats and “spontaneous arrivals” (as noted by several of the same authors in 2006) to the point where now few “immigrated cats” replace cats that die or are removed for adoption [25].

It was suggested by those interviewed for this study that the UCF TNR program’s low profile, including the placement of feeding stations at inconspicuous locations, which allowed cats to wait for food and eat out of public view, was an important factor in reducing abandonment on campus. The university’s ongoing commitment to a “no pets” policy was thought to have discouraged abandonment as well. Limiting the abandonment of pet cats and vigilance in monitoring of the program area for new arrivals are credited with helping to sustain the reductions in free-roaming cat numbers in the long term [26]; this corroborates one of the central findings of stochastic simulation modeling recently published by Boone et al. [8].

### 4.2. Disposition

The various outcomes experienced by cats in the program (i.e., remained on campus, adopted, relocated to the woods, disappeared, died, or euthanized), when expressed as a percentage of the total number of cats enrolled, changed little from 2002 to 2019 (Table 2). Two notable exceptions were the number of cats remaining on campus, which fell from 15% of the total to 5% and the number of cats who disappeared, which increased from 15% to 24%. The fates of cats who disappeared are unknown; however, likely outcomes for these cats include relocation, death by traumatic injury (e.g., vehicle collision), and undeclared adoption by university staff. Survey data indicates that 21% of pet cats in the U.S. are acquired directly from the “stray” population [27]; therefore, some cats who disappeared likely were taken in as pets without the knowledge of colony caretakers.

When the outcomes over the combined observation period are compared to the disposition of cats enrolled in two other long-term TNR programs, Chicago and Sydney, Australia, a smaller percentage of cats remained on site (5% compared to Chicago, 23%, and Sydney, 12%) and disappeared (24% compared to Chicago, 34%, and Sydney, 29%), a greater percentage of cats were adopted (45% compared to Chicago, 30%, and Sydney, 27%), and the percentage of cats euthanized (11% compared to Chicago, 3%, and Sydney, 17%) or known to have died (8% compared to Chicago, 7%, and Sydney, 12%) fell in between what occurred at the other two locations [19,20]. Of note, if the number of cats who moved to the woods were instead categorized as having disappeared, the percentage of cats who disappeared, 29%, would then be identical to what was documented in Sydney. Of note, asymptomatic cats testing positive for FIV and FeLV were routinely euthanized as part of the UCF program from 1991 to 2000; this policy was adhered to early on as part of the Sydney program [19], but was not part of the Chicago program [20]. During the follow-up observation period, two cats were returned to the UCF campus after testing positive for FIV, one, who arrived in 2017, remained on site at the 2019 follow-up, while the other, who arrived in 2016, disappeared three months after his return to colony.

Levy et al. reported that at the end of the initial observation period, “in general, the cats were in adequate physical condition, and only 4% were euthanized for humane reasons” [11]. Through the end of the combined observation period, the percentage of cats euthanized for serious illness or injury increased to only 5%, a strong indication that the general health of the cats on campus remained good. Generally, it also continued to be the case that cats survived for a number of years on campus, as illustrated by the fact that cats not adopted within six months of enrollment remained on campus for more than four years, on average. No attempt was made to approximate the specific age of cats categorized as adults upon enrollment to the program; therefore, the actual lifespan of each of these cats is unknown (but was at minimum six months longer than the amount of time such cats spent on campus).

As noted by Levy et al. [11], adoptions continued to account for a substantial portion of the decrease in the cat population, even among cats categorized as feral. Overall, adoption as a final outcome declined from 47% at the end of the initial observation period to 45% in 2019; however, the adoption of feral cats rose from comprising 58% of total adoptions in 2002 to 62% in 2019. Overall, 65% of the feral cats adopted were kittens. Reasons for the relatively large number of feral cats adopted from the program include concerted efforts by Friends of Campus Cats to socialize feral kittens [26] and as observed elsewhere [13,22], the adoption by colony caretakers of feral adults who grew more sociable over time [26]. As the number of cats on campus declined, so did the average number of cats enrolled in the program each year (falling from an average of 14 per year during the initial observation period to three per year during the follow-up period). Hence, new arrivals were sterilized and when appropriate, adopted from the program more quickly. The overall rate of adoption from UCF’s TNR program was greater than what was reported by long-term programs elsewhere: Newburyport ~33% [22], Chicago, 30% [20], Key Largo, 28% [21], and Sydney, 27% [19].

### 4.3. Population Dynamics

A long-term commitment to the sterilization of the UCF community cat population resulted in a sharp decline in the number of kittens on campus. The percentage of cats categorized as kittens at inclusion in the program decreased from 56% during the 11-year initial observation period to 12% during the 17-year follow-up observation period. All six of the kittens enrolled between 2002 and 2019 were feral immigrants. The last kitten known to have been born on campus was in 1995, four years after program initiation. This is consistent with what has been observed at sites of other long-term TNR programs, where kitten births were eliminated after two to eight years [19,20,22]. In all, kittens made up just over half of the total number of feral cats and 28% of the total number of socialized cats enrolled in the program. The percentage of cats estimated to be under six months of age upon disposition from the program held constant at 14%, and all such cats were removed for adoption; kittens enrolled during the follow-up observation period were quickly placed into homes, typically within a month of discovery.

Levy et al. [11] concluded that the welfare of the cats on the UCF campus was enhanced by the prevention of kitten births, and noted that no kittens were born on campus after 1995. The fact that the campus had remained free of kittens born on site at the end of the combined observation period in 2019, further supports this conclusion. Nutter et al. [28] reported that free-roaming female cats produce a mean of 1.4 litters per year, with a median of three kittens per litter (range, 1–6), and that expected kitten mortality before six months of age is as high as 75%. Consequently, as a result of UCF’s long-term TNR program, hundreds of kitten births were prevented, which likely avoided considerable suffering [25,28] and minimized preventable deaths [8], and along with adoption, made possible the significant and sustained decline of the community cat population over time.

### 4.4. Colony Changes

From 2002 to 2019, the total number of colonies discovered on the UCF campus rose from 11 to 16; however, over the same period, the number of colonies eliminated increased from 3 to 11. The proportion of colonies eliminated exceeds what was observed in an urban Chicago neighborhood, where 8 of 20 colonies were eliminated over a 10-year period [20]. It also exceeds the proportion of feeding stations (individual colonies were not tracked) eliminated over a 14-year period (41 of 85) in Key Largo [21]; however, in Newburyport a higher proportion of feeding stations (13 of 14) were eliminated over a 17-year period [22]. Ongoing surveillance and management of TNR program areas are attributed with preventing the repopulation of disbanded colonies [19,22]. The process of trapping, sterilizing, and often rehoming new arrivals happened more quickly as the overall number of community cats on the UCF campus declined. The significant decrease in the average amount of time between enrollment and sterilization that occurred during the follow-up observation period (Figure 3) is illustrative of this change.

Of the 24 cats who moved from one colony to another, 88% (21/24) were sterilized at the time of their first colony change; however, two male cats, who changed colonies once, went unsterilized (one for 1.8 years and the other for 7.5 years) before disappearing, and another male cat changed colonies three times (including once to the NFC group for a period of 2.9 years) before being sterilized after 7.7 years on campus (and disappearing 2.3 years later). Gunther et al. observed significantly higher rates of emigration among unneutered cats [29]; the aforementioned migration results appear to be inconsistent with those findings. The colony dynamics at play regarding the 41 cats who spent some or all of their time on campus with no fixed colony attachment were not studied; therefore, it is unclear whether this was a matter of preference for these cats or if they were prevented from joining or staying in existing colonies during these periods. Future research into these issues is warranted.

## 5. Study Limitations

The limitations of the present study include those commonly encountered when conducting a retrospective investigation. The absence of certain data (e.g., estimated ages for cats at entry rather than simple categorizations of “kitten” or “adult”) constrained the types of analyses that could be performed. The inability to examine all of the same documents used to calculate results at the end of the initial observation period caused a nominal degree of uncertainty in the reconciliation of some details. Moreover, UCF’s TNR program was set up without control group colonies to allow for the comparison of results. Colony control groups, as established by Nutter [13], offer obvious benefits to researchers, but are unlikely to be integrated into efforts initiated by community-based TNR groups whose primary objective is reducing colony size as quickly as possible.

## 6. Conclusions

In their 2003 paper, Levy et al. [11] concluded that based upon the results attained on the UCF campus, “long-term reduction of free-roaming cat numbers is feasible by TNR.” Based upon a growing body of evidence, including the present study, it appears that long-term reductions of free-roaming cat populations by TNR, of sufficient intensity [8], are not only feasible, but with continued colony management, sustainable over extended periods.

The present study reveals that free-roaming cat numbers on the UCF campus have declined significantly for nearly three decades since the inception of the TNR program. During the follow-up observation period, on average, only three new arrivals were enrolled in the program per year, even as the UCF campus grew into the largest public university (by enrollment for the 2018–2019 academic year) in the U.S. [30]. The durability of the UCF program, despite the university’s steep growth in size over many years, suggests that such results may be sustainable, with proper ongoing management, for an indefinite period and over a variety of contexts.

As suggested by Levy et al. [11], the pairing of adoption with TNR, as was done for the duration of the UCF program, is consistent with conventional animal welfare values. In fact, adoption is now commonly considered part of TNR program’s best practices [31], in large part, because it is effective at expediting reductions in free-roaming cat numbers [14,19,20,22,32,33] when employed as a sensible complement to the central component of TNR, sterilization.

## Figures and Tables

**Figure 1 animals-09-00768-f001:**
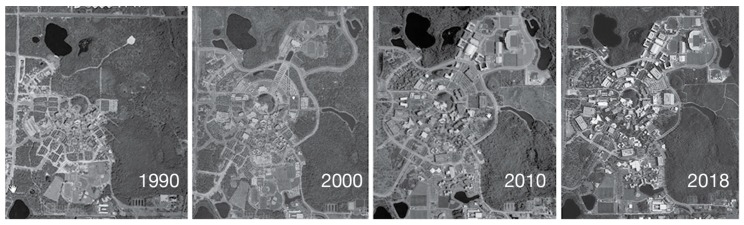
Satellite images of the University of Central Florida (UCF) campus.

**Figure 2 animals-09-00768-f002:**
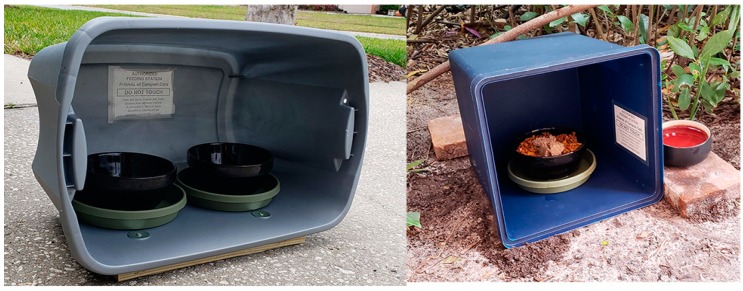
Feeding stations.

**Figure 3 animals-09-00768-f003:**
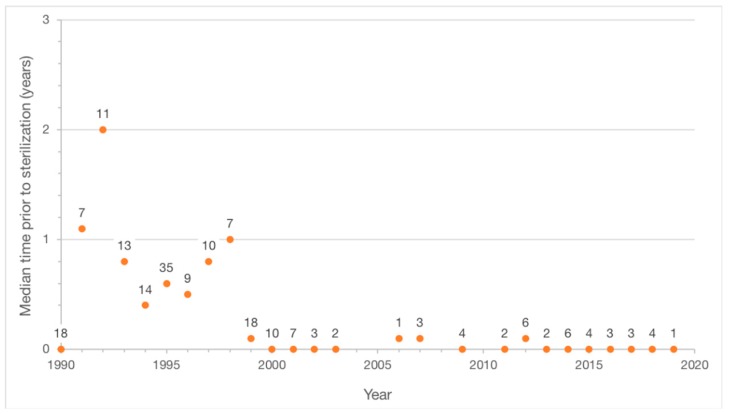
Amount of time between enrollment in a university campus TNR program and sterilization surgery during a 28-year observation period.

**Figure 4 animals-09-00768-f004:**
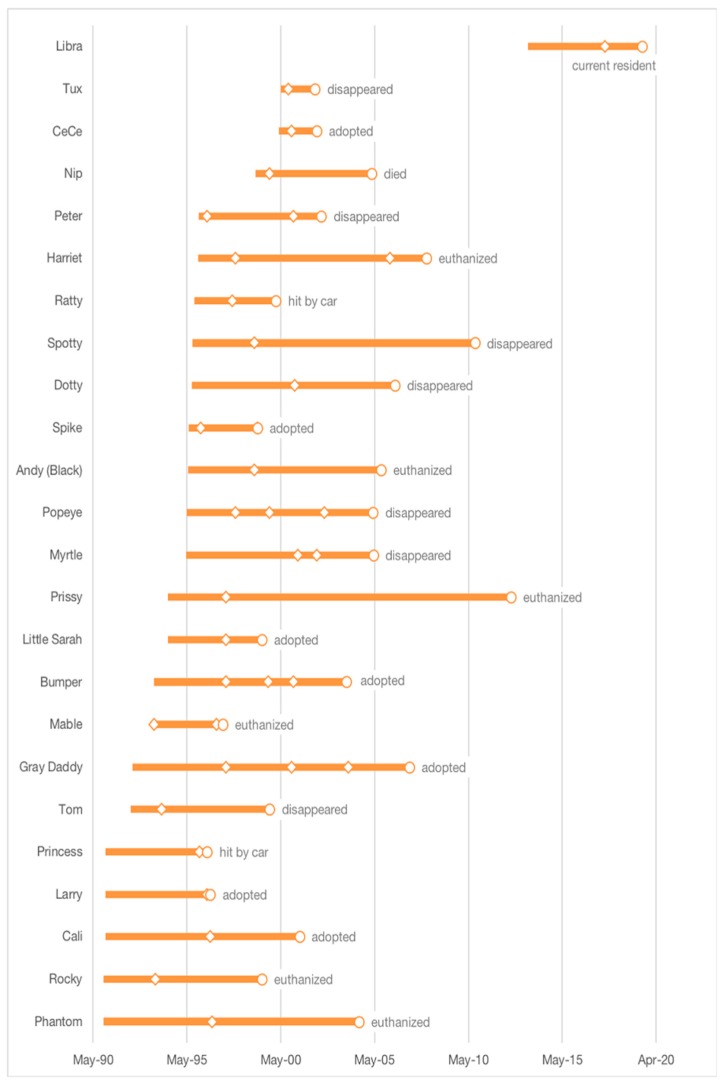
Timing of arrival and disposition of cats who changed colonies over the 28-year observation period. Diamonds indicate time of colony change(s); circles indicate time of disposition.

**Figure 5 animals-09-00768-f005:**
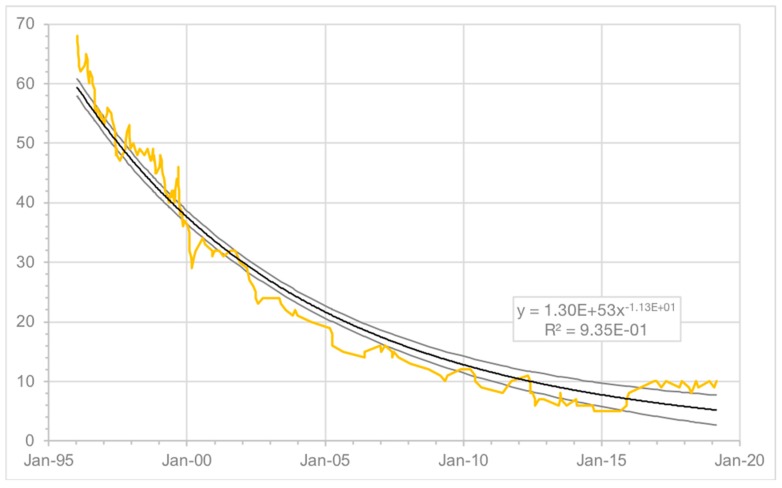
Population of cats documented on UCF campus, 1996–2019 (power fit and 95% confidence intervals shown in gray).

**Table 1 animals-09-00768-t001:** Characteristics of 204 free-roaming cats at inclusion in a trap-neuter-return (TNR) program.

Variable	Feral	Socialized	Total
Population	161	43	204
Age			
Kittens	81	12	93
Adults	80	31	111
Sex			
Male	92	22	114
Female	69	21	90

**Table 2 animals-09-00768-t002:** Disposition of 204 free-roaming cats included in a university campus TNR program.

Disposition	No. of Cats (%)	Sex		Socialization		Age		Time on Campus (Years)		
		M	F	Feral	Sociable	Kitten	Adult	Mean ± SD	Median	Range
Remaining	10 (5%)	3	7	10	0	0	10	3.5 ± 3.1	3.0	0.3–10.0
Adopted	92 (45%)	45	47	57	35	28	64	2.1 ± 3.4	0.3	0–16.3
To woods	10 (5%)	7	3	10	0	0	10	1.5 ± 2.6	0.3	0–9.0
Out of area	3 (1%)	2	1	3	0	0	3	0	-	-
Disappeared	49 (24%)	29	20	48	1	0	49	4.4 ± 4.2	3.1	0–15.2
Died	17 (8%)	11	6	16	1	0	17	3.8 ± 2.6	3.9	0.3–8.3
Euthanized	23 (11%)	17	6	17	6	0	23	4.6 ± 4.4	4.3	0–13.6
Total	204	114	90	161	43	28	176	3.1 ± 3.8	1.6	0–16.3

**Table 3 animals-09-00768-t003:** Adoptions at the end of the initial observation and combined observation periods.

Adoptions; Age and Level of Socialization	2002		2019	
	Total on Campus	No. Adopted (%)	Total on Campus	No. Adopted (%)
Socialized				
Adults	27	19 (70)	31	23 (74)
Kittens	12	12 (100)	12	12 (100)
Feral				
Adults	41	9 (22)	80	20 (25)
Kittens	75	33 (44)	81	37 (46)
Total	155	73 (47)	204	92 (45)

**Table 4 animals-09-00768-t004:** Comparison of results from long-term TNR studies.

Program Location (Source)	University of Central Florida	Newburyport, Massachusetts (22)	Key Largo, Florida (21)	Chicago, Illinois (20)	Sydney, Australia (19)
Duration (years)	28	17	14	4–10	9
Cat population					
Total managed	204	~340	2529	195	122
Initial census	68	~300	455	75†	69
Remaining cats	10	0	206	44	15
(%)	5	0	8	23	12
Population reduction (%)	85	100	55	41	78
Colonies eliminated vs. total	11/16	13/14‡	41/85‡	8/20	NR
Disposition:					
Adoption (%)	45	~33	28 ^	30	27
Disappeared (%)	24	NR	NR	34	29
Euthanized (%)	11	~5–10	17 ^	3	17
Died (%)	8	NR	11 ^	7	12

^†^ Total at entry for all colonies; ^‡^ Feeding stations; ^^^ Outcomes at last recorded veterinary visit; NR = not reported.

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
