# Peer review of "Back to School: An Updated Evaluation of the Effectiveness of a Long-Term Trap-Neuter-Return Program on a University’s Free-Roaming Cat Population"

_animals, 2019, doi:10.3390/ani9100768_

Round 1
Reviewer 1 Report
Whilst this paper is of an important topic and presents important data, it is written and presented so very one-sided that it becomes hard to take seriously. The authors are so very clearly passionate about TNR that they have not taken any time to cross reference themselves or consider alternative viewpoints. There is nothing wrong with passion and believing what you present, but at least some level of cross-checking yourself is required. The introduction does not even mention some of the issues with trap-neuter-release, and reads essentially as TNR is a great method and always works. But what are some of the critiques of the method? And if it is already established as effective, why bother presenting this data? With no good description of the issues, this whole paper just reads like a grab-bag of numbers with no point.
Most of the findings, especially relating to cat population size, will need to be presented with statistics. How do we know the reduction is not related to changes in detectability, or within a margin of error?
Even if the paper is updated and the framing is improved, it also has aloyt of unimportant data that will need to be removed. For example, figure 4 presents all the different names, fates and times of individual cats. I cant see any scenario that this would be important to any reader.
That all said, the story itself of a successful longterm TNR effort is important to present, and i hope a more objective and robust manuscript could be created.
Reviewer 2 Report
This is a great study that adds to the body of literature related to TNR. With 28 years worth of data, publication of this study is important/significant.
A few items deserve clarification.
Even though the authors refer the initial study and the combined study occasionally the reader may get lost in determining which stage of the study is actually being discussed or from which stage data is being presented. Clarification is in order. Is the use of the term "colony" when referring to one cat appropriate? The focus on only 10 cats remaining out of a total of 204 in the study is a little misleading as an additional 10 relocated to the adjacent woods and 3 relocated out of the area. Perhaps some attention should be given to the fact that 20 (or 23 depending on what exactly "out of the area" meant) cats remained as community cats. In the discussion, lines 345 & 346 it states: "Ongoing surveillance and management at former colony sites is attributed with preventing repopulation." Please explain - I don't understand what that sentence is trying to say.Author Response
Response to Reviewer 2 Comments
This is a great study that adds to the body of literature related to TNR. With 28 years worth of data, publication of this study is important/significant.
A few items deserve clarification.
Point 1: Even though the authors refer the initial study and the combined study occasionally the reader may get lost in determining which stage of the study is actually being discussed or from which stage data is being presented. Clarification is in order.
Response 1: Per your suggestion, to add clarity we have revised our language at multiple points in the manuscript (lines 158-159 and 175-176).
Point 2: Is the use of the term "colony" when referring to one cat appropriate?
Response 2: We appreciate your question. We know of no established alternative threshold regarding the number of cats that constitute a colony. The colonies on the UCF campus were dynamic; some increasing in size from a single cat to multiple cats over time, and others decreasing in size from multiple cats to one cat. In addition, the term “colony” is sometimes defined in local ordinances as “one or more cats.”
Point 3: The focus on only 10 cats remaining out of a total of 204 in the study is a little misleading as an additional 10 relocated to the adjacent woods and 3 relocated out of the area. Perhaps some attention should be given to the fact that 20 (or 23 depending on what exactly "out of the area" meant) cats remained as community cats.
Response 3: Cats that relocated to the woods are not known to remain on campus…essentially they disappeared, but were reported in this manner (“to woods”) by Levy et al. in 2003. We indicated the following on lines 153-154 “10 (5%) relocated to one of the wooded areas that surround much of the campus (status unknown)…” Moreover, 9 of the 10 cats had relocated to the woods by 2002, so they are most likely dead and, therefore, should not be added to the total number of cats who currently remain on campus. As is also stated in the text, the tenth cat who moved to the woods (to an adjacent research park) has not been seen since. We included the following in the Discussion: “Of note, if the number of cats who moved to the woods were instead categorized as having disappeared, the percentage of cats who disappeared, 29% (59/204), would then be identical to what was documented in Sydney.”
The three cats who moved “Out of Area” are known to now live outside the boundaries of the program area. So, 10 cats remain in the program area, but three more are known to continue to roam outdoors. We have added the following to the text (in parentheses) on lines 244-245: “A total of 10 cats remained on campus (as stated above, an additional three individuals were known to have moved outside of the program area where they continued to live as community cats), just 5% of the 204 total cats enrolled over the entirety of the program and an 85% decline from the 68 cats present in 1996 (Figure 5).”
Point 4: In the discussion, lines 345 & 346 it states: "Ongoing surveillance and management at former colony sites is attributed with preventing repopulation." Please explain - I don't understand what that sentence is trying to say.
Response 4: We have reworded the sentence in question (now at lines 361-362) in hopes of making it clearer: “Ongoing surveillance and management of TNR program areas are attributed with preventing the repopulation of disbanded colonies [12,15].”
It is, of course, possible that we are misunderstanding your concerns. If that is the case, please say so and we'll make every effort to better address them. We appreciate your feedback!
Round 2
Reviewer 1 Report
The steps taken to address my previous comments were in the right direction, but I am really advocating that a major overhaul in writing style and presentation is needed.
To re-clarify my comments on the first draft, essentially the gist of your previous introduction was TNR always works. If so, then there is no point in presenting your data. But of course that is not how this issue is in reality, and there is much debate about it. It is not a one-sided “the sky is blue” style issue like you presented, but complex and has many viewpoints (as you clearly knew). You have now made some steps to improve the introduction and give some background to the points of contention, but not enough.
A new problem now is you use terms like “failed to eliminate scepticism…” and “critics point to…”. This is not a bloody editorial, it is a scientific paper. Discuss what the actual gap in the literature is. Instead of partisan comments like that, at least make some effort to sound like you are objective here. Might I suggest something along the lines of “whilst some studies of TRN have not reported in a decline in cat numbers, these are typically short term. More long-term studies are needed”. Then, bookend with a discussion on the importance of long-term monitoring, and suggest the short term studies are flawed.
Your actual dataset here seems solid and important, but you do it a disservice by being so one-sided and blinkered in your reporting of it.
Also, you need to add some sentence or two on why you even want to manage and control cat populations in the first place.
In regards to statistics and figures, this manuscript presents things is so different from what I though was the accepted norm, that I am going to have to step back and let the associate editor handle this. My understanding is that you should avoid using raw numbers in the discussion as much as possible (you put them in almost every sentence), that * denotes significance (you have it denote estimate, which I thought was ~), that everything on a graph needs to be labelled or described in text (e.g. what are the triangles in fig 4), that graphs should maximise the ink to information ratio (you have background lines, un-necessary labels, ect), and that you should only report an increase or decline using statistics (you consistently just report the stand-alone figures). Take this sentence below from your discussion; “adoption as a final outcome declined only slightly from 47% at the end of the initial observation period to 45% in 2019;” Without statistics, how do we know what you report is a slight decline or just natural variation? But maybe this journal is different from the others I am familiar with.
And one more comment, the following line from your discussion (350) is so strange to me I don’t really know how to process it or provide comment “Consequently, as a result of UCF’s long-term TNR program, hundreds of kitten births were prevented, which likely averted considerable suffering [18,22] and preventable death [20],”. I feel like the logical extension here would be that killing adult sea turtles is good as it alleviates suffering of many more baby sea turtle deaths. There are many reasons I can think of that reducing kitten births of feral cats is positive and the program presented here is a success story, but this makes no sense to me. Maybe just me though.
